# Relationship between Nutrition Intake and 28-Day Mortality Using Modified NUTRIC Score in Patients with Sepsis

**DOI:** 10.3390/nu11081906

**Published:** 2019-08-15

**Authors:** Dae Hyun Jeong, Sang-Bum Hong, Chae-Man Lim, Younsuck Koh, Jarim Seo, Younkyoung Kim, Ji-Yeon Min, Jin Won Huh

**Affiliations:** 1Department of Internal Medicine, University of Ulsan College of Medicine, Asan Medical Center, 88, Olympic-ro 43-gil, Songpa-gu, Seoul 05505, Korea; 2Division of Pulmonary & Critical Care Medicine, University of Ulsan College of Medicine, Asan Medical Center, 88, Olympic-ro 43-gil, Songpa-gu, Seoul 05505, Korea; 3Department of Pharmacy, University of Ulsan College of Medicine, Asan Medical Center, 88, Olympic-ro 43-gil, Songpa-gu, Seoul 05505, Korea; 4Food and Nutrition Service Team, University of Ulsan College of Medicine, Asan Medical Center, 88, Olympic-ro 43-gil, Songpa-gu, Seoul 05505, Korea

**Keywords:** energy, modified NUTRIC score, mortality, protein, sepsis

## Abstract

In critically ill patients, malnutrition is known to increase morbidity and mortality. We investigated the relationship between nutritional support and 28-day mortality using the modified NUTrition RIsk in the Critically ill (NUTRIC) score in patients with sepsis. This retrospective cohort study included patients with sepsis admitted to the medical intensive care unit (ICU) between January 2011 and June 2017. Nutritional support for energy and protein intakes at day 7 of ICU admission were categorized into <20, 20 to <25, and ≥25 kcal/kg and <1.0, 1.0 to <1.2, and ≥1.2 g/kg, respectively. NUTRIC scores ≥4 were considered to indicate high nutritional risk. Among patients with low nutritional risk, higher intakes of energy (≥25 kcal/kg) and protein (≥1.2 g/kg) were not significantly associated with lower 28-day mortality. In patients with high nutritional risk, higher energy intakes of ≥25 kcal/kg were significantly associated with lower 28-day mortality compared to intakes of <20 kcal/kg (adjusted hazard ratio (aHR): 0.569, 95% confidence interval (CI): 0.339–0.962, *p* = 0.035). Higher protein intakes of ≥1.2 g/kg were also significantly associated with lower 28-day mortality compared to intakes of <1.0 g/kg (aHR: 0.502, 95% CI: 0.280–0.900, *p* = 0.021). Appropriate energy (≥25 kcal/kg) and protein (≥1.2 g/kg) intakes during the first week may improve outcomes in patients with sepsis having high nutritional risk.

## 1. Introduction

Malnutrition is very common among critically ill patients, with a prevalence of 20–50% [1,2,3,4]. Malnourished patients have increased rates of morbidity and mortality [5,6,7,8,9]. The leading causes of mortality in the intensive care unit (ICU) are sepsis and septic shock. Owing to exaggerated acute phase responses and gastrointestinal dysfunction, patients with sepsis have hypermetabolism, which results in a greater risk of malnutrition [10]. Guidelines for nutritional support in patients with sepsis are based on findings of studies among heterogenous groups of critically ill patients [11,12].

In recent years, many studies have indicated the importance of protein in nutrition support [13]; optimal protein intakes potentially lower mortality in critically ill patients [14,15]. As per the proceedings of the International Protein Summit, high doses of protein may be required in ICU settings to optimize nutritional support and improve mortality rates [16]. Certain guidelines recommend a protein intake of 1.2–2 g/kg/day for patients with sepsis [12]. A prospective observational study found that the achievement of both, protein and energy targets was associated with decreased 28-day mortality. However, solely achieving the energy target was not found to be associated with reduced mortality [14].

The findings of studies on the impact of energy intake in critically ill patients have been conflicting. Two randomized controlled trials comparing underfeeding with standard feeding in critically ill patients showed no difference in clinical outcomes [17,18]. Arabi et al. compared outcomes between groups receiving permissive underfeeding and standard feeding; the findings showed no difference in 90-day mortality in a subgroup of 292 patients with sepsis [18]. A previous study has demonstrated that near-target caloric intakes were associated with adverse outcomes, including increased hospital mortality and infection-related complications [19]. Conversely, hypocaloric nutrition has been associated with an increased prevalence of nosocomial infections in critically ill patients [20]. In an international multicenter observational study, greater protein and energy intakes, using the modified NUTrition RIsk in the Critically ill (NUTRIC) score, were found to be associated with lower mortality and shorter times to discharge alive in high-risk patients [21].

In view of inconsistencies from previous reports, we hypothesized that the underfeeding in critically ill patients with hypermetabolic status may reach to the suboptimal intake of energy and protein, leading to poor prognosis. This retrospective study was done to investigate the relationship between nutritional support during the first week and 28-day mortality, using the modified NUTRIC score in patients with sepsis.

## 2. Materials and Methods

### 2.1. Study Participants

We retrospectively analyzed the data of a prospectively collected sepsis cohort in the medical ICU of the Asan Medical Center, a tertiary referral hospital in Seoul, South Korea. Patients with sepsis, aged at least 18 years, who stayed in the ICU for more than 7 days, between January 2011 and June 2017, were included in the study.

#### Ethical Statement

The study protocol was approved by the Institutional Review Board of the Asan Medical Center (No 2017-0833). The requirement for informed consent was waived owing to the retrospective nature of the study.

### 2.2. Data Collection

We reviewed patient data from electronic medical records, and collected information pertaining to demographics, height, weight, comorbidities, diagnosis, length of stay (LOS) in the ICU, mechanical ventilation (MV), vasopressor drug use, and renal replacement therapy (RRT). Scores were calculated using data from the first 24 h after ICU admission. We calculated the modified NUTRIC scores (0–9) using the available data [22]. Scores ≥4 were considered to be indicative of high nutritional risk [23] and were classified as high-risk. Scores <4 were considered to indicate low nutritional risk. Energy (25–30 kcal/kg/day) and protein (1.2–1.5 g/kg/day) requirements were calculated using a simple weight-based equation [12]. The achieved daily energy and protein intakes were calculated by combining intakes from enteral nutrition (EN) and parenteral nutrition (PN) for days 1, 3, and 7. We followed a nutrition support protocol, which stipulated that the nutrition intake within 1 week of initiation of nutritional support should achieve 20–25 kcal/kg/day and 1.0–1.2 g/kg/day for energy and protein, respectively. The categories for nutritional adequacy at day 7 of ICU admission were as follows: <20 kcal/kg, 20 to <25 kcal/kg, and ≥25 kcal/kg for energy, and <1.0 g/kg, 1.0 to <1.2 g/kg, and ≥1.2 g/kg for protein. This categorization was selected to ensure that nutritional intake was provided correctly according to our nutrition support protocol, and to assess 28-day mortality according to nutritional adequacy.

### 2.3. Statistical Analysis

We compared high and low nutritional risk patients using the modified NUTRIC score. Categorical and continuous variables were compared using the chi-square and Student’s *t*-tests, respectively. The Cox proportional hazards model was used to evaluate the outcome variable, namely, 28-day mortality. Hazard ratios (HR) were adjusted for age, sex, BMI (Body Mass Index), co-morbidities, and energy or protein intake. All tests of significance were 2 sided; a *p*-value < 0.05 was considered statistically significant. All statistical analyses were performed using the SPSS software (Version 21.0; SPSS Inc., Chicago, IL, USA) package.

## 3. Results

### 3.1. Patients Characteristics

The characteristics of the 248 patients included in this study, based on the risk group allocated using the modified NUTRIC score, are shown in Table 1. Overall, 30.2% of patients were female, the median age of patients was 68 (interquartile range (IQR): 57–74) years, and the median body mass index (BMI) was 22 (IQR: 19–25) kg/m^2^. The median Acute Physiology and Chronic Health Evaluation (APACHE) II score, sequential organ failure assessment (SOFA) score, and the median number of comorbidities were 22 (IQR: 19–28), 11 (IQR: 8–14), and 2 (IQR: 1–3), respectively. A total of 107 patients (43.1%) had neoplasms. The median LOS in the ICU was 13 days (IQR: 9–25 days), and MV, vasopressor drugs, and RRT was required in 217 (87.5%), 226 (91.1%), and 104 (41.9%) patients, respectively. The overall 28-day mortality was 34.3% (17.9% and 36.4% in patients with low and high NUTRIC scores, respectively).

### 3.2. Nutritional Profiles

The nutritional profiles of patients, based on the modified NUTRIC score, are shown in Figure 1. The mean energy target at the end of week 1 was 1526 kcal/day, and the mean achieved energy intakes on days 1, 3, and 7 were 36.1%, 77.2%, and 86.2% of the energy target, respectively. At the end of week 1, the mean protein target was 71 g/day, and the mean achieved protein intakes on days 1, 3, and 7 were 31.9%, 70.6%, and 79.7% of the protein target, respectively. The rate of achievement of protein targets was relatively low compared with that of energy targets. On an average, in the groups at low and high nutritional risk, EN was initiated on days 3.0 and 4.3 of admission to the ICU, respectively. On day 7, 25.0% and 26.4% of patients at low and high risk, respectively, received nutritional support in the form of EN, while 50% and 40%, received a combination of EN and PN, respectively.

### 3.3. Nutritional Intakes and 28-Day Mortality

The relationship between nutritional intake and 28-day mortality is shown in Table 2. In the group at high nutritional risk, the 28-day mortality tended to decrease with an increase in the intakes of energy or protein. Table 3 shows the HR for 28-day mortality in the entire cohort. Higher energy intake was significantly associated with lower 28-day mortality in those with intakes of ≥25 kcal/kg than those taking <20 kcal/kg (20 to <25 kcal/kg: Adjusted hazard ratio (aHR): 0.883, 95% confidence intervals (CI): 0.513–1.520, *p* = 0.654; ≥25 kcal/kg: aHR: 0.534, 95% CI: 0.322–0.887; *p* = 0.015). Higher protein intake was significantly associated with lower 28-day mortality in those taking ≥1.2 g/kg than those with intakes of <1.0 g/kg (1.0 to <1.2 g/kg: aHR: 0.860, 95% CI: 0.476–1.555, *p* = 0.618; ≥1.2 g/kg: aHR: 0.475, 95% CI: 0.270–0.836, *p* = 0.010).

Table 4 shows the HR for 28-day mortality in the group at low nutritional risk. Among patients with low nutritional risk (<4 points), higher energy intake was not significantly associated with lower 28-day mortality in the group with intakes of 20 to <25 and ≥25 kcal/kg compared to that with <20 kcal/kg. Higher protein intake was also not significantly associated with lower 28-day mortality.

Table 5 shows the HR for 28-day mortality in the group at high nutritional risk. For patients with high nutritional risk (≥4 points), higher energy intake was significantly associated with lower 28-day mortality in the group taking ≥25 kcal/kg compared to that with intakes of <20 kcal/kg (aHR: 0.569, 95% CI: 0.339–0.962, *p* = 0.035). In addition, higher protein intakes were significantly associated with lower 28-day mortality in the group with intakes of ≥1.2 g/kg compared to that taking <1.0 g/kg (aHR: 0.502, 95% CI: 0.280–0.900, *p* = 0.021).

## 4. Discussion

In our study, achieving the nutritional target was associated with lower mortality in patients with sepsis. In the group at high nutritional risk, meeting the nutritional goal was associated with lower mortality. However, in the group at low nutritional risk, the impact of higher nutritional intakes, in terms of achievement of the energy or protein targets, was not observable. In the high-risk group, the rates of achievement of the protein targets were significantly lower than that of the energy targets (78.3% vs. 84.8%, *p* < 0.001). This finding suggests that it is necessary to monitor the achievement of protein targets within at least one week in critically ill patients with sepsis, particularly in those at high nutritional risk.

Although guidelines suggest that EN should be initiated early within the first seven days in critically ill patients with sepsis [11], the use of EN was relatively low in the high-risk group. Although EN plays a major role, it may be difficult to achieve the nutritional targets with EN alone. In our study, the feeding routes at day 7 were EN, EN combined with PN, and PN in 26.2%, 41.1%, and 32.7% of patients, respectively. In the NUTRIREA-2 study, clinical outcomes including mortality and the risk of ICU-acquired infections in critically ill patients with shock did not differ between those receiving early EN and PN [24]. As mentioned previously, improving nutritional intake, irrespective of the feeding route may improve clinical outcomes including mortality, nosocomial infections, and quality of life. In our study, on day 7, high risk patients received about 13% and 12% less energy and protein, respectively, than low risk patients. High risk patients received more parenteral energy and protein than low risk patients. As per our nutritional support protocol, supplemental PN was provided in cases where EN was inadequate. Heidegger et al. showed that the optimization of energy provisions with supplemental PN could reduce nosocomial infections among patients with critical illnesses [25]. In our cohort, higher nutrition intake was significantly associated with lower mortality in high risk patients. This suggests that adequate nutrition through supplemental PN may improve clinical outcomes in critically ill patients with sepsis.

Conversely, near-target energy intakes have been associated with increased hospital mortality and ICU-acquired infections [19]. A randomized controlled trial reported that in critically ill patients, permissive underfeeding may be associated with lower mortality than target feeding [26]. However, in another trial, hypocaloric feeding in a subgroup of 292 patients with sepsis showed no difference in 90-day mortality compared to standard feeding [18]. In the TARGET trial, increasing the energy intake with energy-dense EN did not impact survival among critically ill patients when protein intake was kept constant. On subgroup analysis, in which 55% of patients had sepsis, the risk of death by day 90 did not differ significantly [27]. In contrast, since the increased energy intake correlated with the increase in protein intake in our study, underfeeding up to day 7 was associated with higher mortality. In addition, the mean BMI of the patients in our study was 22 kg/m^2^, which was relatively low compared to those of previous studies; this may explain our findings. In a majority of previous studies, the mean patient BMI was in excess of 25 kg/m^2^. Higher energy or protein intake has been associated with lower mortality in patients with a BMI of <25 and ≥35 kg/m^2^; no benefits are reported in patients with a BMI of 25 to <35 kg/m^2^ [28]. In the present study, most patients (73.8%) had a BMI of less than 25 kg/m^2^. Another possible explanation for the observed beneficial impact of higher nutritional intake in our cohort, is that we only included patients with sepsis who were in a state of catabolic stress. A multicenter cohort study in critically ill patients requiring prolonged MV reported that higher nutritional intakes during the first week in the ICU were associated with longer survival and shorter times to physical recovery. Wei et al. therefore suggested that current recommendations to underfeed critically ill patients may cause harm in certain patients who stayed longer in the ICU [29]. Although the effect of underfeeding remains unclear in critically ill patients, underfeeding in patients who stay in the ICU for long periods, such as our study population, require particular attention.

In recent years, in addition to overall energy intake, the importance of providing protein has gained importance in the feeding of critically ill patients. Allingstrup et al. showed that increased protein intake was associated with lower mortality, with 10-day survival rates of 50%, 78%, and 87% in groups with low (0.79 g/kg), medium (1.06 g/kg), and high (1.46 g/kg) protein intake, respectively [15]. Similarly, we found that an increased protein intake resulted in a stepwise decline in mortality (39.3%, 31.1%, and 25.4% in group with <1.2, 1.0 to 1.2, and ≥1.2 g/kg, *p* = 0.048). Nicolo et al. demonstrated that achieving at least 80% of the protein target was associated with lower mortality and a shorter time to discharge alive in ICU patients [30]. Similarly, in our study, the group taking ≥1.2 g/kg of protein had significantly lower mortality compared to that taking <1.0 g/kg. Weijs et al. reported that early high protein intake was associated with lower mortality in critically ill patients without sepsis; however, there was no beneficial effect on mortality among patients with sepsis [31]. Unlike our study, that study included only 117 patients (14%) with sepsis, and protein intake was only analyzed on day 4.

Owing to the differences in conditions and comorbidities between patients, the response to nutritional intervention is not always uniform. Patients with sepsis who are malnourished or at risk of malnutrition may receive the greatest benefit from nutritional support [32]. Previous studies have indicated that the beneficial effect of appropriate nutritional support is more apparent in patients at high-nutritional risk [28,33], with higher nutritional intake being associated with lower mortality in patients with high NUTRIC scores [21,34,35]. Similarly, in our study, a higher nutritional intake was associated with lower mortality in patients at high nutritional risk. On post hoc analysis of the PermiT trial, in which 33% of patients had sepsis, permissive underfeeding provided similar outcomes to standard feeding in both, high- and low-risk patients [36]. These results may have been due to similar protein intakes in each group. Therefore, the intakes of both energy and protein should be considered while establishing nutritional goals. Further studies are needed to determine the appropriate amounts of protein required in patients in the ICU, stratified by nutritional risk.

Our study has several limitations. First, this was a single center retrospective study; therefore, the compliance with feeding and the cause of interruption of nutritional support may not have been correctly documented. Second, other confounders, such as disease severity and pre-existing malnutrition, which may have influenced the results were not considered. Third, the commercial enteral or parenteral nutrition formulations used in the ICU have relatively low protein content (40–48 g/L); therefore, reaching the target protein intake without protein supplementation was difficult. Fourth, our study was conducted in an exclusively Asian cohort. Generalization of these results to Western populations with different BMI distributions may not be feasible. However, Compher et al. have reported that the benefit of greater protein/energy intake in patients with high NUTRIC scores was observed irrespective of geographic origin [37].

## 5. Conclusions

Our results suggest that achieving the nutritional goal, including both energy and protein, within the first week of nutritional support may improve 28-day mortality in patients with sepsis having high nutritional risk. In addition to macronutrients, recent data have reported that probiotics, or pharmaconutrients such as vitamin D and selenium, affect the clinical outcome of sepsis [32]. Further multicenter prospective studies are needed to evaluate the adequacy of nutritional support considering the supportive therapy such as pharmaconutrients or probiotics in patients with sepsis having high nutritional risk.

## Figures and Tables

**Figure 1 nutrients-11-01906-f001:**
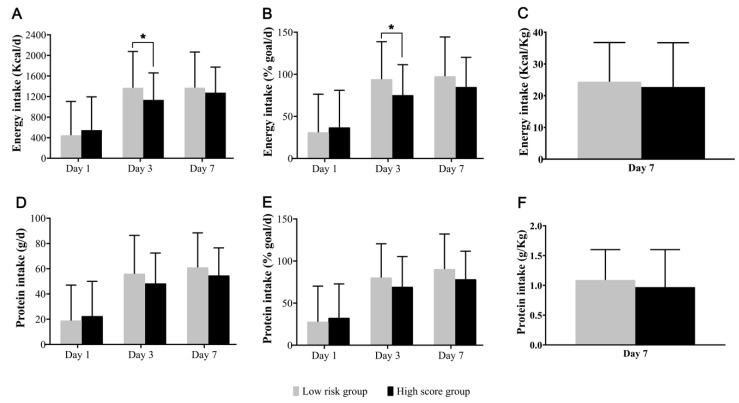
Energy and Protein Intake according to modified NUTRIC score. (**A**,**D**) show the total amounts of energy and protein received on day 1, 3, and 7. (**B**,**E**) show the percentage of reaching amounts in whom the target of 25 kcal (energy) and 1.2 g (protein) per kilogram of body weight per day was achieved. (**C**,**F**) show the total amounts of energy and protein received per kilogram of actual or estimated body weight at day 7; data are presented as means and standard deviation. * *p*-value < 0.05).

**Table 1 nutrients-11-01906-t001:** Patient characteristics according to modified NUTRIC score.

Variable	Modified NUTRIC Score (*n* = 248)		
	Low Score	High Score	*p*-Value
	(*n* = 28)	(*n* = 220)	
Age, years	54 (42–70)	68 (58–75)	0.001
Height, cm	164 (160–170)	163 (155–170)	0.375
Weight, kg	57 (49–61)	60 (51–67)	0.548
BMI, kg/m^2^	21 (18–24)	23 (19–25)	0.156
Female, n (%)	7 (25.0)	68 (30.9)	0.521
APACHE II score	15 (13–18)	24 (20–28)	<0.001
SOFA score	6 (4–8)	12 (9–14)	<0.001
Days from hospital to ICU	0 (0–2)	0 (0–6)	0.510
Co-morbidities	1 (1–2)	2 (1–3)	0.013
LOS in ICU, days	11 (9–19)	14 (9–25)	0.586
MV	21 (75.0)	196 (89.1)	0.034
Vasopressor use	22 (78.6)	204 (92.7))	0.013
RRT	4 (14.3)	100 (45.5)	0.002
Diagnosis			0.434
Respiratory disease	19 (62.9)	119 (54.1)	
Liver/GI disease	3 (10.7)	35 (15.9)	
Cardiovascular disease	0 (0)	8 (3.6)	
Renal disease	1 (3.6)	14 (6.4)	
Febrile neutropenia	0 (0)	11 (5.0)	
SSTI	0 (0)	8 (3.6)	
Other	5 (17.9)	25 (11.4)	
VAP	0 (0)	19 (8.6)	0.106
Bacteremia	8 (28.6)	77 (35.0)	0.500
Sepsis severity			<0.001
Sepsis	0 (0)	6 (2.7)	
Severe sepsis	19 (67.9)	65 (29.5)	
Septic shock	9 (32.1)	149 (67.7)	
28-day mortality	5 (17.9)	80 (36.4)	0.052
Sepsis related deaths	4 (80)	40 (50)	0.193

Data are presented as number (%) or median (IQR). NUTRIC, NUTrition RIsk in the Critically ill; APACHE, Acute Physiology and Chronic Health Evaluation; BMI, Body Mass Index; CRP, C-reactive protein; GI, gastrointestinal; ICU, intensive care unit; LOS, length of stay; MV, mechanical ventilation; RRT, renal replacement therapy; SOFA, Sequential Organ Failure Assessment; SSTI, skin and soft tissue infection; VAP, ventilator associated pneumonia.

**Table 2 nutrients-11-01906-t002:** Relation between nutrition intake and 28-day mortality.

	Low Score (*n* = 28)			*p*-Value	High Score (*n* = 220)			*p*-Value
Energy intake (kcal/kg)	<20	20 to <25	≥25		<20	20 to <25	≥25	
No. of patients	10	3	15		90	53	77	
Deaths	3 (30.0)	0 (0)	2 (13.3)	0.323	39 (43.3)	19 (35.8)	22 (28.6)	0.048
Protein intake (g/kg)	<1.0	1.0 to <1.2	≥1.2		<1.0	1.0 to <1.2	≥1.2	
No. of patients	12	5	11		128	40	52	
Deaths	3 (25.0)	1 (20.0)	1 (9.1)	0.330	52 (40.6)	13 (32.5)	15 (28.8)	0.117

Data are presented as number (%).

**Table 3 nutrients-11-01906-t003:** Cox regression analysis for 28-day mortality in total study population.

Variables	Unadjusted		Adjusted	
	HR (95% CI)	*p*-Value	HR (95% CI)	*p*-Value
Age	1.008 (0.992–1.015)	0.334	1.015 (0.998–1.032)	0.091
Sex	1.823 (1.177–2.826)	0.007	2.009 (1.285–3.140)	0.002
BMI	1.051 (1.007–1.098)	0.024	..	..
Co-morbidities	1.054 (0.849–1.309)	0.631	..	..
Energy intake (kcal/kg)				
<20	Reference		Reference	
20 to <25	0.865 (0.503–1.487)	0.599	0.883 (0.513–1.520)	0.654
≥25	0.562 (0.340–0.929)	0.025	0.534 (0.322–0.887)	0.015
Protein intake (g/kg)				
<1.0	Reference		Reference	
1.0 to <1.2	0.793 (0.441–1.427)	0.643	0.860 (0.476–1.555)	0.618
≥1.2	0.506 (0.289–0.886)	0.017	0.475 (0.270–0.836)	0.010

BMI, Body Mass Index; HR, Hazard ratios.

**Table 4 nutrients-11-01906-t004:** Cox regression analysis for 28-day mortality in low nutritional risk group.

Variables	Unadjusted		Adjusted	
	HR (95% CI)	*p*-Value	HR (95% CI)	*p*-Value
Age	0.993 (0.954–1.034)	0.742		
Sex	1.001 (0.274–3.657)	0.999		
BMI	1.050 (0.931–1.184)	0.424	1.081 (0.927–1.260)	0.320
Co-morbidities	0.565 (0.296–1.078)	0.083		
Energy intake (kcal/kg)				
<20	Reference		Reference	
20 to <25	0 (0)	0.989	0 (0)	0.988
≥25	0.400 (0.062–2.563)	0.334	0.193 (0.013–2.840)	0.231
Protein intake (g/kg)				
<1.0	Reference		Reference	
1.0 to <1.2	0.806 (0.079–8.187)	0.855	0.021 (0.000–9.343)	0.214
≥1.2	0.285 (0.028–2.909)	0.289	0.256 (0.013–5.176)	0.375

BMI, Body Mass Index.

**Table 5 nutrients-11-01906-t005:** Cox regression analysis for 28-day mortality in high nutritional risk group.

Variables	Unadjusted		Adjusted	
	HR (95% CI)	*p*-Value	HR (95% CI)	*p*-Value
Age	1.009 (0.990–1.028)	0.345		
Sex	1.977 (1.235–3.165)	0.005	1.970 (1.239–3.132)	0.004
BMI	1.049 (1.001–1.099)	0.044		
Co-morbidities	1.117 (0.884–1.411)	0.354		
Energy intake (kcal/kg)				
<20	Reference		Reference	
20 to <25	0.864 (0.499–1.495)	0.601	0.901 (0.520–1.561)	0.709
≥25	0.582 (0.344–0.982)	0.043	0.569 (0.337–0.962)	0.035
Protein intake (g/kg)				
<1.0	Reference		Reference	
1.0 to <1.2	0.780 (0.425–1.432)	0.423	0.857 (0.464–1.583)	0.622
≥1.2	0.537 (0.301–0.956)	0.035	0.502 (0.280–0.900)	0.021

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
