# Peer review of "Relationship between Nutrition Intake and 28-Day Mortality Using Modified NUTRIC Score in Patients with Sepsis"

_nutrients, 2019, doi:10.3390/nu11081906_

Round 1
Reviewer 1 Report
It is an interesting retrospective study on the rather underestimated issue on the effect of malnutrition and nutritional support on the outcome of critically ill patients with sepsis.
However more details should be given on the issue of the infection details of the patients
How many patients had VAP ? bacteremias ? etc...
Were there any differences between groups in relation to the sepsis severity (sepsis vs severe sepsis vs septic shock) ?
Was the final outcome directly related to sepsis (sepsis related deaths) or other comorbidities ?
Author Response
It is an interesting retrospective study on the rather underestimated issue on the effect of malnutrition and nutritional support on the outcome of critically ill patients with sepsis.
However more details should be given on the issue of the infection details of the patients
How many patients had VAP ? bacteremias ? etc...
Were there any differences between groups in relation to the sepsis severity (sepsis vs severe sepsis vs septic shock) ?
Was the final outcome directly related to sepsis (sepsis related deaths) or other comorbidities ?
-> Thank you for your comment.
We added the clinical information.
In high nutritional risk group, the proportion of patients with septic shock was significantly higher compared to low nutritional risk group.
The incidence of VAP or bacteremia was not significantly different between two groups
.
We added in the revised table 1 as follows:
Revised manuscript (Results, Pages 3~4, lines 114)
|
Variable |
Modified NUTRIC score (n = 248) |
||
|
Low score |
High score |
P-value |
|
|
(n = 28) |
(n = 220) |
||
|
Age, years |
54 (42-70) |
68 (58-75) |
0.001 |
|
Height, cm |
164 (160-170) |
163 (155-170) |
0.375 |
|
Weight, kg |
57 (49-61) |
60 (51-67) |
0.548 |
|
BMI, kg/m2 |
21 (18-24) |
23 (19-25) |
0.156 |
|
Female, n (%) |
7 (25.0) |
68 (30.9) |
0.521 |
|
APACHE II score |
15 (13-18) |
24 (20-28) |
<0.001 |
|
SOFA score |
6 (4-8) |
12 (9-14) |
<0.001 |
|
Days from hospital to ICU |
0 (0-2) |
0 (0-6) |
0.510 |
|
Co-morbidities |
1 (1-2) |
2 (1-3) |
0.013 |
|
LOS in ICU, days |
11 (9-19) |
14 (9-25) |
0.586 |
|
MV |
21 (75.0) |
196 (89.1) |
0.034 |
|
Vasopressor use |
22 (78.6) |
204 (92.7)) |
0.013 |
|
RRT |
4 (14.3) |
100 (45.5) |
0.002 |
|
Diagnosis |
0.434 |
||
|
Respiratory disease |
19 (62.9) |
119 (54.1) |
|
|
Liver/GI disease |
3 (10.7) |
35 (15.9) |
|
|
Cardiovascular disease |
0 (0) |
8 (3.6) |
|
|
Renal disease |
1 (3.6) |
14 (6.4) |
|
|
Febrile neutropenia |
0 (0) |
11 (5.0) |
|
|
SSTI |
0 (0) |
8 (3.6) |
|
|
Other |
5 (17.9) |
25 (11.4) |
|
|
VAP |
0 (0) |
19 (8.6) |
0.106 |
|
Bacteremia |
8 (28.6) |
77 (35.0) |
0.500 |
|
Sepsis severity |
|
|
<0.001 |
|
Sepsis |
0 (0) |
6 (2.7) |
|
|
Severe sepsis |
19 (67.9) |
65 (29.5) |
|
|
Septic shock |
9 (32.1) |
149 (67.7) |
|
|
28-day mortality |
5 (17.9) |
80 (36.4) |
0.052 |
|
Sepsis related death |
4 (80) |
40 (50) |
0.193 |
In addition, we analyzed these clinical data in high-nutritional risk group according to energy or protein intake. (Data not presented in the revised manuscript)
|
|
High score (n = 220) |
P-value |
||
|
Energy intake (kcal/kg) |
<20 |
20 to <25 |
≥25 |
|
|
No. of patients |
90 |
53 |
77 |
|
|
Sepsis severity |
|
|
|
0.016 |
|
Sepsis |
1 (1.1) |
2 (3.8) |
3 (3.9) |
|
|
Severe sepsis |
21 (23.3) |
15 (28.3) |
29 (37.7) |
|
|
Septic shock |
68 (75.6) |
36 (67.9) |
45 (58.4) |
|
|
VAP |
8 (8.9) |
6 (11.3) |
5 (6.5) |
0.605 |
|
Bacteremia |
30 (33.3) |
26 (49.1) |
21 (27.3) |
0.470 |
|
Deaths |
39 (43.3) |
19 (35.8) |
22 (28.6) |
0.048 |
|
Sepsis related deaths |
19 (48.7) |
11 (57.9) |
10 (45.5) |
0.896 |
|
Protein intake (g/kg) |
<1.0 |
1.0 to <1.2 |
≥1.2 |
|
|
No. of patients |
128 |
40 |
52 |
|
|
Sepsis severity |
|
|
|
0.694 |
|
Sepsis |
3 (2.3) |
1 (2.5) |
2 (3.8) |
|
|
Severe sepsis |
38 (29.7) |
11 (27.5) |
16 (30.8) |
|
|
Septic shock |
87 (68.0) |
28 (70.0) |
34 (65.4) |
|
|
VAP |
9 (7.0) |
6 (15.0) |
4 (7.7) |
0.654 |
|
Bacteremia |
45 (35.2) |
16 (40.0) |
16 (30.8) |
0.686 |
|
Deaths |
52 (40.6) |
13 (32.5) |
15 (28.8) |
0.117 |
|
Sepsis related deaths |
25 (48.1) |
8 (61.5) |
7 (46.7) |
0.888 |

Reviewer 2 Report
Interesting paper focused on nutritional status assessment in patients with sepsis.
Some minor issues are:
the absence of hypotheses about the effect of underfeeding in critically ill patients (except a fair consideration at line 210 and subseq.)
in despite of the good text conclusions are almost inconsistent and must be expanded and made more interesting for the readers, also including some consideration about current supportive therapies (i.e. probiotics)
Author Response
Interesting paper focused on nutritional status assessment in patients with sepsis.
Some minor issues are:
the absence of hypotheses about the effect of underfeeding in critically ill patients (except a fair consideration at line 210 and subseq.)
-> Thank you for your comment.
We revised the manuscript as follows:
Revised manuscript (Introduction, Pages 2, lines 63-65)
In view of inconsistencies from previous reports, we hypothesized that the underfeeding in critically ill patients with hypermetabolic status may reach to the suboptimal intake of energy and protein, leading to poor prognosis. This retrospective study was done to investigate the relationship between nutritional support during the first week and 28-day mortality, using the modified NUTRIC score in patients with sepsis.
.
in despite of the good text conclusions are almost inconsistent and must be expanded and made more interesting for the readers, also including some consideration about current supportive therapies (i.e. probiotics)
-> Thank you for your comment.
We agree with the reviewer’s opinion. In addition to macronutrients, recent data showed that pharmaconutrients such as vitamin D and selenium or probiotics is promising. But, the response of supportive therapies such as pharmaconutrients and probiotics is not always uniform. The effect of supportive therapies may be different by underlying condition or nutritional risk status
We revised the sentence as follows:
Revised manuscript (Conclusion, Pages 8, lines 251-256)
Our results suggest that achieving the nutritional goal, including both, energy and protein, within the first week of nutritional support may improve 28-day mortality in patients with sepsis having high nutritional risk. In addition to macronutrients, recent data have reported that probiotics, or pharmaconutrients such as vitamin D and selenium, affect the clinical outcome of sepsis. Further multicenter prospective studies are needed to evaluate the adequacy of nutritional support considering the supportive therapy such as pharmaconutrients or probiotics in patients with sepsis having high nutritional risk.
